# Selective Removal of Chlorophyll and Isolation of Lutein from Plant Extracts Using Magnetic Solid Phase Extraction with Iron Oxide Nanoparticles

**DOI:** 10.3390/ijms25063152

**Published:** 2024-03-09

**Authors:** Jolanta Flieger, Natalia Żuk, Sylwia Pasieczna-Patkowska, Marcin Kuśmierz, Rafał Panek, Wojciech Franus, Jacek Baj, Grzegorz Buszewicz, Grzegorz Teresiński, Wojciech Płaziński

**Affiliations:** 1Department of Analytical Chemistry, Medical University of Lublin, Chodźki 4A, 20-093 Lublin, Poland; natalia.zuk@umlub.pl; 2Department of Chemical Technology, Faculty of Chemistry, Maria Curie Skłodowska University, Pl. Maria Curie-Skłodowskiej 3, 20-031 Lublin, Poland; sylwia.pasieczna-patkowska@mail.umcs.pl; 3Analytical Laboratory, Institute of Chemical Sciences, Faculty of Chemistry, Maria Curie12 Skłodowska University, Pl. Marii Curie-Skłodowskiej 3, 20-031 Lublin, Poland; marcin.kusmierz@mail.umcs.pl; 4Department of Geotechnics, Civil Engineering and Architecture Faculty, Lublin University of Technology, Nadbystrzycka 40, 20-618 Lublin, Poland; r.panek@pollub.pl (R.P.); w.franus@pollub.pl (W.F.); 5Department of Forensic Medicine, Medical University of Lublin, 20-090 Lublin, Poland; jacek.baj@umlub.pl (J.B.); grzegorz.buszewicz@umlub.pl (G.B.); grzegorz.teresinski@umlub.pl (G.T.); 6Department of Biopharmacy, Medical University of Lublin, 4a Chodźki Str., 20-093 Lublin, Poland; wojciech.plazinski@umlub.pl; 7Jerzy Haber Institute of Catalysis and Surface Chemistry, Polish Academy of Sciences, Niezapominajek Str., 30-239 Krakow, Poland

**Keywords:** chlorophyll a, iron oxide nanoparticles, lutein, magnetic solid phase extraction, photosynthetic pigments

## Abstract

In recent years, there has been a growing interest in plant pigments as readily available nutraceuticals. Photosynthetic pigments, specifically chlorophylls and carotenoids, renowned for their non-toxic antioxidant properties, are increasingly finding applications beyond their health-promoting attributes. Consequently, there is an ongoing need for cost-effective methods of isolation. This study employs a co-precipitation method to synthesize magnetic iron oxide nanoparticles. Scanning electron microscopy (SEM) coupled with energy dispersive spectrometry (EDS) confirms that an aqueous environment and oxidizing conditions yield nanosized iron oxide with particle sizes ranging from 80 to 140 nm. X-ray photoelectron spectroscopy (XPS) spectra indicate the presence of hydrous iron oxide FeO(OH) on the surface of the nanosized iron oxide. The Brunauer–Emmett–Teller (BET) surface area of obtained nanomaterial was 151.4 m^2^ g^−1^, with total pore volumes of pores 0.25 cm^3^ g^−1^ STP. The material, designated as iron oxide nanoparticles (IONPs), serves as an adsorbent for magnetic solid phase extraction (MSPE) and isolation of photosynthetic pigments (chlorophyll a, lutein) from extracts of higher green plants (*Mentha piperita* L., *Urtica dioica* L.). Sorption of chlorophyll a onto the nanoparticles is confirmed using UV–vis spectroscopy, Fourier transform infrared photoacoustic spectroscopy (FT-IR/PAS), and high-performance liquid chromatography (HPLC). Selective sorption of chlorophyll a requires a minimum of 3 g of IONPs per 12 mg of chlorophyll a, with acetone as the solvent, and is dependent on a storage time of 48 h. Extended contact time of IONPs with the acetone extract, i.e., 72 h, ensures the elimination of remaining components except lutein, with a spectral purity of 98%, recovered with over 90% efficiency. The mechanism of chlorophyll removal using IONPs relies on the interaction of the pigment’s carbonyl (C=O) groups with the adsorbent surface hydroxyl (–OH) groups. Based on molecular dynamics (MD) simulations, it has been proven that the selective adsorption of pigments is also influenced by more favorable dispersion interactions between acetone and chlorophyll in comparison with other solutes. An aqueous environment significantly promotes the removal of pigments; however, it results in a complete loss of selectivity.

## 1. Introduction

Commonly available and inexpensive green plants have gained considerable attention in the food, pharmaceutical, and cosmetic industries due to their nutritional value and associated health benefits. Extracts of these plants contain significant amounts of photosynthetic pigments, particularly chlorophylls and carotenoids. These pigments are known for their antioxidant, anti-mutagenic, anti-viral, anti-inflammatory, and anti-cancer effects [1,2,3]. In addition, plants have been explored as potential sources of chlorophylls and other photosynthetic pigments for the conversion of solar energy into electricity, particularly in third-generation photovoltaic cells known as dye-sensitized solar cells [4].

Chlorophylls, considered nutraceuticals, have been linked to potential benefits such as blood sugar lowering, detoxification, improved digestion, and reduced allergen levels. In the cosmetic and food industries, chlorophylls are used as green pigments and odor-absorbing additives, as seen in E140 chlorophyll and E141 chlorophyllin, the sodium–copper salt of chlorophyll [4]. Carotenoids, a diverse group of over 1000 plant pigments including β-carotene, lutein, and zeaxanthin, are known for their beneficial effects in preventing eye diseases such as cataracts, age-related macular degeneration (AMD), and retinal degeneration [5,6,7]. Recently, attention has focused on the protective role of carotenoids against ultraviolet A (UVA) radiation and skin damage [8,9,10,11,12]. Lutein is also registered as nutraceutical in DrugBank (Accession Number DB00137). There are two pharmaceutical products containing lutein in a form of capsules and powder for oral administration, and a few mixture and unapproved products. Furthermore, lutein is registered as one of the food additives.

Obtaining high-purity natural pigments is a costly endeavor and requires selective and highly efficient isolation methods [13]. The use of solvent extraction by acetone or water–ethanol causes strong coloration of green plant extracts. Many efforts have been made to dechlorophyllize plant extracts or to isolate coexisting photosynthetic pigments in high purity.

Plant extracts can be subjected to chromatographic analysis to isolate chlorophylls and carotenoids [14,15,16]. In this case, there is a high risk of degradation of pigments, especially chlorophyll. Classic extraction methods include liquid–liquid extraction (LLE), also known as solvent extraction [17], and its version for ionic analytes, ion pair extraction (IPE) [18]. Despite satisfactory extraction efficiency, these methods do not meet the criteria of so-called green chemistry due to the high consumption of toxic organic solvents [19].

Dispersive liquid–liquid microextraction (DLLME) solves this problem by using much smaller amounts of organic solvent [20,21]. However, DLLME is mainly useful for the purification of aqueous solutions from various trace contaminants. One of the more environmentally friendly versions of DLLME is its use in deep eutectic solvent (DES) extraction [22]. Another extraction method is liquid–solid extraction (LSE) [21] or solid phase extraction (SPE) [23]. This process of separating the analyte from the sample matrix and interfering compounds involves analyte–adsorbent interactions of various types, such as ion exchange, physical adsorption, hydrophobic effect, etc. On an analytical scale, this is a widely used method, but on a macro scale, it is not very profitable due to the high cost of adsorbents for solid phase extraction.

The solution to this problem is dispersive solid phase extraction (d-SPE), in which a sorbent is added to the liquid sample. After selective adsorption of the analyte, it is recovered with a solvent [24]. The latest version of d-SPE is magnetic solid phase extraction (MSPE). The advantage of this method is that a sorbent with magnetic properties is dispersed in the sample solution. This provides a large interfacial contact area and the ability to separate the sorbent from the solution using an external magnetic field [25,26].

Various methods for the removal of green pigments from plant extracts have been investigated, including liquid–solid extraction using polymers and molecularly imprinted polymers [27,28]. In addition, novel technologies using aqueous solutions of ionic liquids, palm oil, non-ionic surfactants, and magnetic nanoparticles modified with an amine group have been proposed [29,30,31,32,33]. Ferreira et al. proposed in 2021 a technology for the extraction of chlorophyll from biomass using aqueous solutions of ionic liquids (ILs), in particular aqueous solutions of hexadecylpyridinium chloride ([C16py]Cl) [29]. Extraction of chlorophyll from biomass using palm oil [30] or non-ionic surfactants [32] has also been proposed. There are several patents for the removal and simultaneous destruction of pigment by radiation (Korean Patent No. 10-1256222) and the removal of pigment together with other active ingredients using diatomaceous earth (Korean Patent No. 10-1288303). In the case of lutein, an isolation method using magnetic nanoparticles modified with an amine group (FluidMAG-Amine) is known [31]. Removal of chlorophyll, carotenoids, and sterols from food and plant tissue samples is possible using commercial SPE BEKOlut^®^ CARBON GCB columns. There are works in the literature that use iron oxide nanoparticles modified with chlorophyll. The synthesized nanocomposites are based on surface-modified nanoparticles, e.g., with silica, e.g., Fe_3_O_4_/SiO_2_ [32] or trimethylammonium hydroxide [34]. Nanomaterials containing metal oxides, particularly magnetite (Fe_3_O_4_) or maghemite (γ-Fe_2_O_3_), have gained widespread interest due to their high affinity for many chemical compounds, small size and high surface area to volume ratio, homogeneity of structure, and superparamagnetism, which allows phase separation by an external magnetic field [35,36,37,38,39].

Carotenoids and chlorophylls coexist with each other in green plant extracts. To isolate them in pure form, multi-stage purification processes should be used, which are not economical on a larger scale. On the other hand, the chemical synthesis of natural pigments is a multi-step, time-consuming process requiring many organic substances as starting substrates. That is why new environmentally friendly cheap methods that can be used on an industrial scale to isolate lutein from natural plant sources are constantly being sought.

The aim of this work is to provide a cost-effective method for the removal of chlorophyll a (Figure 1) from plant extracts using magnetic solid phase extraction (MSPE), employing biosorbent in the form of iron oxide nanoparticles (IONPs), and to achieve efficient isolation/extraction of lutein (Figure 2). The conditions for extraction of chlorophyll a and isolation of lutein from extracts of higher green plants, such as *Urtica dioica* L. and *Mentha piperita* L., were optimized. The methodology described in this manuscript has been granted a Polish patent (P.447168, 19 December 2023) entitled “Method for selective removal of chlorophyll a from the extract of green plants and isolation of lutein from the extract of green plants”.

## 2. Results

### 2.1. Characteristics of IONPs Obtained by Co-Precipitation Method

#### 2.1.1. SEM-EDS

Scanning electron microscopy (SEM) was utilized to depict the morphology of IONPs. The SEM images of IONPs are depicted in Figure 3. The predominant shapes of IONPs were spherical, with noticeable aggregation of nanoparticles. Energy dispersive spectrometry (EDS) analysis indicated signals at 6.4 and 7.1 keV, attributed to iron (Fe).

#### 2.1.2. XPS

XPS analysis was used to chemically characterize the IONP surface. Figure 4 shows a fitted XPS spectrum of the Fe2p region. The peak set used to recreate the spectrum envelope was based on the model described by Grosvenor [42].

It follows relative binding energies, peak widths, and intensities of four main components (peaks 1–4). The numerical parameters of fitting of the Fe2p spectrum with FeOOH peaks, i.e., binding energies (BE), peak widths (eV), and their intensities (%) based on the model published by Grosvenor [42], are presented in Table 1. However, the peak position is slightly shifted toward higher binding energies, and the fit clearly indicates a presence of FeOOH on the surface.

Additionally, we fitted the spectra using two models by Biesinger [43]. The first consists of peaks corresponding to FeOOH, and the second includes peaks for FeOOH and FeO. The two fitting results are presented in Appendix A. They are on par with the very first one presented in Figure 4, confirming the presence of FeOOH, or at least Fe (III); however, the second one does not exclude a tiny amount of FeO. For the sake of simplicity, the composite is termed as iron oxide nanoparticles (IONPs) in this work.

### 2.2. Characterization of IONPs Using the Brunauer–Emmett–Teller (BET) Analysis

The BET-specific surface area as well as the pore size and the radius were determined based on the vapor nitrogen adsorption/desorption isotherm at −195.843 °C. Figure 5 shows the N_2_ sorption–desorption isotherms of IONPs.

According to recommendations of the International Union of Pure and Applied Chemistry (IUPAC), the obtained shape of isotherm can be classified as IV (a) type of BET isotherms [44]. As can be seen from the graph, in the low-pressure region (P/P_0_ < 0.4), the adsorption and desorption isotherms superposition. At the region of P/P_0_ > 0.4, the isotherms increase rapidly and form a lag loop. This shape reflects the phenomena of adsorption in micropores at lower pressures and subsequent capillary agglomeration as it grows.

In order to calculate the specific surface area, a linear graph with a high regression coefficient was made, showing the relationship P/P_0_/n(1 − P/P_0_) versus P/P_0_. Figure 6 shows the BET line plot for IONPs.

The specific surface area calculated using the BET equation was 151.377 ± 1704 m^2^ g^−1^. Single-point total pore volumes of pores less than 20.7391 nm diameter at pressure P/P_0_ = 0.9 were 0.2367 cm^3^ g^−1^ and 0.2509 cm^3^ g^−1^ for adsorption and desorption processes, respectively. The adsorption and desorption cumulative surface areas were calculated not only by BET but also using Langmuir, BJH, and single-point methods. Table 2 summarizes the results of experiments for the tested nanoparticles. The pores’ properties, calculated by BJH, such as the cumulative pore-specific surface area, the cumulative pore volume, and the average pore diameter, are collected in Table 3.

The Barrett–Joyner–Halenda (BJH) analysis of the pore size distribution in IONPs is shown in Figure 7. As can be seen, the pore size distribution lies in d > 2 nm indicating a mesoporous structure. The average pore diameter is approximately 6 nm. However, the pore size distribution determined based on the desorption of nitrogen presented in Figure 7 indicates the predominance of pores in the range of 9.1 to 12.3 nm

### 2.3. Optimizing Chlorophyll a Extraction from Plant Extracts

Figure 8 illustrates the change in the absorption intensity of the acetone extract of *Urtica dioica* L. after treatment with IONPs. Chlorophyll, as a pigment, has two distinct absorption bands. The first (420–430 nm), located in the blue-violet region (the Soret band), is a characteristic of all porphyrin derivatives. The second, located at 660 nm (the Q band) in the red region of the spectrum, is typical of compounds derived from dihydroporphin, such as chlorophyll. The spectrum obtained, therefore, represents the specific form of the chlorophyll absorption spectrum, with an absorption gap in the green region. It is noticeable that after two hours of contact with IONPs, the intensity of the absorption peak at λ_max_ = 423 nm significantly decreased, and the Q band at 662 nm almost disappeared. Carotenoids, typically masked by chlorophyll, became visible (see Figure 8, insert), giving the supernatant a yellow color, together with three characteristic absorption bands located in the blue-violet part of the spectrum. Changes in the extract spectrum after nanoparticle addition are accompanied by a visible color shift from green to yellow, which can be seen with the naked eye.

#### 2.3.1. Adsorption Isotherm

By adding 0.15 to 0.5 g of IONPs to 3 mL of dilute acetone extract of *Urtica dioica* L., 35.09% to 96.06% of chlorophyll a was removed from the extract sample. Figure 9 shows the isotherm of the adsorption process of chlorophyll a on IONPs, plotted on the basis of the collected experimental data together with the prediction of the Langmuir adsorption model. Based on the parameter values of the fitted function, the capacity of the monolayer was determined to be 0.1025 mg/mg, and the affinity of chlorophyll for IONPs was 2.707 mL/mg (understood as the value of the Langmuir constant). The applicability of the Langmuir model was very good (R^2^ = 0.9634), suggesting a sorption mechanism based on single-site monolayer adsorption with uniform sorption energy.

#### 2.3.2. The Adsorption Kinetics

A critical property of sorbents is the time required to immobilize the substance to be removed from the solution. To investigate the effect of contact time on the efficiency of chlorophyll a sorption on IONPs, mixtures containing 0.4 g of IONPs and 3 mL of acetone extract, diluted to give an absorbance of approximately 1 at 662 nm (8-fold dilution), were subjected to centrifugation in a rotor for periods ranging from 0 to 2 h.

The kinetic curve illustrating the adsorption efficiency, expressed as % removal of chlorophyll a (*C*_0_ = 0.51125 mg mL^−1^), as a function of contact time for IONPs is shown in Figure 10. The amount of retained chlorophyll a per unit mass of IONPs proportionally increases with increasing contact time. In the initial phase (first 40 min), the adsorption process is rapid, resulting in the retention of 81.17% of the total chlorophyll a content. Subsequently, the adsorption process significantly slows down, reaching an optimal immobilization time of 2 h for the diluted extract (90.84%). Beyond this period, further changes in absorbance are not significant, and keeping the unseparated phases in contact for 24 h does not induce any further changes.

In the case of the undiluted extract, 3.2 g of IONPs was introduced to 3 mL of the extract containing 4.0 mg mL^−1^ of chlorophyll a. Following the time of no less than 48 h, 99.01% of chlorophyll a was successfully extracted.

#### 2.3.3. The Influence of Water on the Sorption Efficiency

The effect of introducing water into the extraction solvent on the chlorophyll removal efficiency of IONPs was investigated in a system containing 3 mL of an 8-fold diluted extract and 0.4 g of IONPs. As shown by the experimental data in Table 4, the changes in the absorbance of the extract (ΔA, %) at two characteristic wavelengths, namely 662 and 452 nm, induced by the addition of IONPs show an increasing trend with contact time. However, the most pronounced changes are observed within the first hour. The presence of more than 30% water in the extraction mixture induces non-selective sorption of all the dyes, including both chlorophyll and carotenoids. The effect of water on the reduced selectivity of photosynthetic pigment sorption by IONPs is illustrated in the photograph in Figure 8. The judicious use of acetone, preferably without the addition of water or with a water content not exceeding 30% (Table 4, Figure 11), ensures the selective removal of chlorophyll from the extract. The observed reduction in absorbance at 662 nm is 90.62% for pure acetone.

### 2.4. Confirmation of Selective Sorption of Chlorophyll a on IONPs

#### 2.4.1. HPLC Analysis

Confirmation of the selective sorption of chlorophyll by IONPs and the synthesis of chlorophyll-modified magnetic iron oxide nanoparticles is supported by HPLC analysis. For this purpose, 3 mL of acetone extract without water containing no more than 4.02 mg mL^−1^ of chlorophyll a was combined with 3.2 g of IONPs, and after 48 h, the nanoparticles were isolated using a magnet. After phase separation, the supernatant was subjected to HPLC analysis. Figure 12 shows the chromatogram of the chlorophyll a standard at a concentration of 3.65 mg mL^−1^ (A), the chromatogram of the extract (B), and the chromatogram of the liquid phase after 48 h contact time with IONPs (C). The chromatograms show the selective removal of chlorophyll under the conditions of analysis.

#### 2.4.2. The FT-IR/PAS Spectra

Figure 13 shows the FT-IR/PAS spectra of chlorophyll, iron oxide nanoparticles (IONPs), and IONPs with adsorbed chlorophyll. The FT-IR spectrum of chlorophyll a shows bands characteristic for the functional groups that build its structure, i.e., C=O, C=C, C–H, and N–H groups. A broad band with two maxima in the range of 3650–3200 cm^−1^ is attributed to the stretching vibrations of the O–H and N–H groups. The bands at 2952, 2924, and 2860 cm^−1^ indicate the presence of –CH_3_ and –CH_2_– groups. A band at 1732 cm^−1^ points to C=O stretching vibrations in the ester group. The band at 1652 cm^−1^ indicates the presence of C=O chlorophyll keto groups [45]. However, this band may also be attributed to C=O vibrations in the carboxylate ion [46] and vibrations of C=C and/or C–N groups [47,48]. The band at 1605 cm^−1^ indicates the presence of C=C bonds, while the band at 1538 cm^−1^ is attributed to C=C and C=N [46] or N–H [48] functional groups. Bands below 1400 cm^−1^ are characteristic for C–O, C–O–C, C–N, and C–H vibrations. The bands in the range of 1000–800 cm^−1^ indicate C–C vibrations in pyrrole rings and the vibrations of C–H groups, while the bands below 800 cm^−1^ are the result of the presence of Mg ions in the chlorophyll structure (Mg–N bond) and the presence of C–N bonds [49]. The band at 1431 cm^−1^ may be the result of the presence of both C–O, C=C groups, and C–H groups [50].

Analysis of the FT-IR/PAS spectrum of IONPs confirms that they are iron oxide nanoparticles [51]. The spectrum shows well-defined peaks at 3439, 1638, 1357, 938, 632, and 541 cm^−1^ and a peak of low intensity at 3695 cm^−1^, which presence indicates the stretching vibrations of the vOH hydroxyl groups. The two peaks at 632 and 570 cm^−1^ result from the presence of iron–oxygen (Fe–O) bonds. The peaks at 3439 and 1638 cm^−1^ are the result of bending vibrations of –OH hydroxyl groups and adsorbed water, respectively. The bands at 1357, 1090, 938, and 824 cm^−1^ indicate the presence of nitrate groups (precursor of iron ions).

The spectrum of IONPs with adsorbed chlorophyll a shows vibration bands of the O–H and N–H groups (3355 and 3300 cm^−1^) and vibrations of the –CH_3_ and –CH_2_– groups (2960–2860 cm^−1^). The bands of C=O groups are slightly shifted compared with the chlorophyll a spectrum (from 1732 to 1701 cm^−1^ and from 1652 to 1638 cm^−1^), which indicates the involvement of these groups in binding with IONPs. In turn, the disappearance of the band at 3695 cm^−1^ in the pristine IONP spectrum indicates the involvement of surface –OH groups of NPs in binding with chlorophyll a. Other bands characteristic of chlorophyll a, which are absent in the spectrum of pristine nanoparticles, also appear in the spectrum. These are the bands at 1431 cm^−1^ (C–O, C=C, C–H), 1370, 1345 cm^−1^ (C–H), 1285, 1237, and 897 cm^−1^ (deformation vibrations of C–H groups in –CH_3_) and bands in the range of 1090–916 cm^−1^ (C–O, C–O–C). The analysis of the FT-IR/PAS spectrum of nanoparticles with adsorbed chlorophyll confirms that chlorophyll has adsorbed on the IONP surface.

### 2.5. Interactions with Solvent

Due to the lack of parameters describing with sufficient accuracy all components of the system (i.e., organic sorbate molecules, solvents, as well as inorganic iron hydroxide with surface hydroxyl groups), MD-based modeling was limited to a series of simplified systems containing only a solute (sorbate) and a solvent, without a sorbent. The results were mainly analyzed in the context of the energy interactions between a solute and a solvent, with only a qualitative analysis of the conformation of solute molecules. In this latter aspect, no significant influence of the solvent type on the conformation of the investigated molecules was observed. The most notable difference is the tendency for closer interaction between the aliphatic chain of chlorophyll and the group containing pyrrole rings, observed in the system containing water. However, due to the fact that chlorophyll adsorption occurs in both solvents, it is difficult to consider this effect significant in the context of adsorption selectivity.

Table 5 presents the average interaction energies between sorbate molecules and the solvent. Interaction energies are always the highest for chlorophyll, regardless of the solvent, which is likely a result of the largest size of the molecule and its greater surface contact with the solvent. Differences in interaction energies between the two solvents indicate that chlorophyll exhibits the greatest asymmetry in the balance of solute–solvent interactions compared with lutein and β-carotene (difference greater by ca. 100 kJ/mol). This is primarily influenced by significantly more favorable dispersion interactions between acetone and chlorophyll, surpassing both electrostatic interactions and analogous terms for the other solutes.

### 2.6. Isolation of Lutein from the Acetone Extract of Urtica dioica L.

About 3.2 g of IONP was added to 3 mL of acetone extracts from *Urtica dioica* L. The samples were placed in a rotor at room temperature, and then the liquid phase was periodically analyzed for dye content. It was established that the most favorable contact time necessary for the isolation of lutein under the conditions of the analysis cannot be shorter than 72 h. Figure 14 shows the chromatograms of the extract before and after sorption with IONPs along with the chromatogram of the lutein standard. Figure 15 shows overlapped spectra, recorded in the range of 350–670 nm by a DAD, for the lutein standard and isolated lutein. The above isolation method ensures the recovery of 99.13% of lutein with a spectral purity of 0.9973.

### 2.7. Applicability of the Procedure of Chlorophyll Removal/Lutein Isolation

#### 2.7.1. *Mentha piperita* L. Acetone Extract

To showcase the versatility of the proposed method for chlorophyll a removal from another green plant, an acetone extract was prepared from dried *Mentha piperita* L. using the same procedure as employed for *Urtica dioica* L. Chromatographic analysis indicated a chlorophyll a content of 3.36 mg mL^−1^ in the extract. Subsequently, 3 mL of the extract was combined with a precisely weighed amount of 3.2 g of IONPs. The samples were analyzed following the identical protocol as for *Urtica dioica* L. The method demonstrated the removal of 99.98% of chlorophyll a and the recovery of 99.13% of lutein with a spectral purity of 0.9471.

#### 2.7.2. *Urtica dioica* L. Dried 80% Ethanol Extract

Green plant extracts can be prepared using alternative solvents. An 80% ethanol extract of *Urtica dioica* L., prepared at room temperature with ultrasound assistance for 1 h, was dried and subsequently dissolved in acetone. Chromatographic analysis revealed a chlorophyll a content of 3.04 mg mL^−1^. A 3 mL aliquot of the extract was mixed with a precisely weighed amount of 3.2 g of IONPs and subjected to the same analytical procedure as before. This approach resulted in the removal of 99.99% of chlorophyll a and the recovery of 96.88% of lutein.

## 3. Discussion

Nanoscale iron oxides significantly differ from their bulk counterparts in terms of optical, electronic, magnetic, and chemical properties [52]. Moreover, they possess a higher surface area and adsorption capacity toward different molecules and ions. In the literature, there are many examples of the use of nanoscale iron oxides as efficient adsorbents suitable for the decontamination of drinking water from heavy metal ions such as Cr(VI), Pb(II), Cr(III), Cu(II), Zn(II), Ni(II), and Cd(II) [53,54] or as potential drug carriers [55,56,57,58].

In this work, IONPs synthesized by co-precipitation were used for the selective removal of chlorophyll from the extract of green plants. The spectra obtained in the SEM-EDS study were similar to those reported in previous works [38,59], which confirms the reliability and repeatability of this synthesis method. The synthesized IONPs had magnetic properties. Under the influence of an applied external magnetic field, they separated from the solution, which ensured an easy separation process. It is known that if magnetic nanoparticles are very small, they may exhibit superparamagnetic properties (40 to 90 Am^2^/kg). In this case, aggregates re-disperse after the magnet removal [60]. In the case of our research, superparamagnetism is excluded by the size of the nanoparticles, which exceeds 100 nm.

It should be noted that the vibrating sample magnetometer (VSM) analysis revealing the magnetic properties of IONPs and the analysis of the surface area and pore size are performed infrequently concerning nanoparticles [61] similar to investigation in the form of differential scanning calorimetry (DSC) and thermogravimetric analysis (TGA) [62]. Waseem et al. [63] found the BET surface area for iron oxide nanoparticles fabricated by a modified hydrolysis technique to be 90.71 m^2^ g^−1^. In turn, the pore diameter for the particles was found to be 10.8 nm, with a pore volume of 0.61 cm^3^ g^−1^. Another study by Mpelane et al. [64] showed a surface area value of 73.6 m^2^ g^−1^ of the mesoporous iron oxide synthesized using a soft-template method. In our case, the BET surface area of the nanomaterial obtained by the co-precipitation method was higher at 151.4 m^2^ g^−1^, with a smaller total pore volume of 0,25 cm^3^ g^−1^.

The analysis of the FT-IR/PAS spectrum confirms that chlorophyll a has adsorbed on the IONP surface through C=O groups. The estimated sorption capacity, which was 102.5 mg g^−1^, is a high value compared with the sorption capacity values estimated for other solid sorbents with respect to chlorophyll [65,66,67,68,69]. So far, the sorption capacities of micro- and mesoporous sorbents have been tested, ranging from 0.3 to 20 mg g^−1^, from minerals (e.g., sepiolite, bentonite, and mesoporous silica) to activated carbon. In our previous work, the sorption capacity of chlorophyll a on polyvinyl chloride (PVC) was 1.12 mg/g [28]. Therefore, IONPs, due to their small size, have an advantage compared with conventional adsorbents due to their larger surface area and available sorption sites.

According to the conducted research, the effectiveness of IONPs in removing chlorophyll and isolating lutein depends on the contact time of the extraction mixture, the water content in the extraction mixture, and the ratio of chlorophyll a (mg) in the extract to the mass of nanoparticles.

Acetone has proven to be the preferred solvent for selective solid phase extraction of chlorophyll, and the water content should not be greater than 30%. A higher water content enhances the hydrophobic effect, which, in turn, may be responsible for the aggregation of chlorophyll a to minimize the interaction of the hydrophobic part of the molecule with water. Other authors also confirm the existence of a strong hydrophobic effect, which is responsible for the self-aggregation of chlorophyll molecules, which increases with time and with the increase in the mole fraction of water [70]. In acetone, the chlorophyll a monomer remains the only species present. This is evidenced by the absence of a bathochromic shift of the red peak associated with the resulting aggregate with an absorption maximum around 713 nm (S713). Thus, the behavior of chlorophyll with respect to IONPs depends on the nature of the organic solvent, especially the water content, which drives the aggregation processes. This approach would suggest that the selectivity of chlorophyll sorption to IONPs occurs only when it is present in the monomeric form. Despite many authors emphasizing the high efficiency of IONP biosorbents for the purification of aqueous solutions from trace organic and inorganic pollutants [71,72,73], it should be emphasized that our study shows that the aquatic environment ensures high sorption efficiency in relation to photosynthetic pigments but does not ensure sorption selectivity.

The developed method of dispersive magnetic extraction ensures selective removal of chlorophyll from plant extracts, the so-called dechlorophyllization. So far, many methods have been used for this purpose using n-hexane, water partitioning, activated charcoal bleaching, and ChloroFiltr^®^ decolorization [74]. Obtaining decolorized extracts can be useful in food and cosmetic industries, etc. The proposed method is, above all, cheaper compared with ready-made ChloroFilt^®^ filters. Compared with solvent extractions, our method ensures lower consumption of organic solvents and the production of an additional product in the form of chlorophyll-modified nanoparticles with the possibility of their further use. Our method is universal. The presented research is based on exemplary acetone extracts of two green plants and dry water–ethanol extract, which is traditionally used to prepare food extracts from plant materials [75]. The problem with these extractions, which our method solves, is that the use of ethanol or acetone results in the co-extraction of high concentrations of chlorophyll [76].

As a result of optimizing the sorption conditions, it was possible to isolate high-purity lutein in a simple and cheap way. It can, therefore, be used on an industrial scale, especially since there is a great demand for lutein obtained from natural sources as a medicinal substance, dietary supplement and a safe, coloring food additive. Chemical synthesis is practically uneconomical and non-ecological.

The weaknesses of the work and the prospect of further research concern a molecular description that would shed light on the causes of sorption selectivity. Despite MD-based modeling revealing the tendency for closer interaction between the aliphatic chain of chlorophyll and the group containing pyrrole rings in water, such an approximate analysis cannot be used to explain the selectivity of chlorophyll adsorption for different solvent compositions due to the lack of a crucial element in the studied systems (i.e., sorbent). However, this approximate analysis indicates significant differences like solute–solvent interactions between chlorophyll and other compounds. Further research is also required to understand sorption kinetics and adsorption isotherms. For this purpose, tests of the specific surface area and pore size of IONP aggregates should be planned. Future research should also include the use of modified IONPs as new biosorbents, dye-sensitized solar cell components, and tests of their biological activity.

## 4. Materials and Methods

### 4.1. Chemicals

Fe(III) chloride hexahydrate (FeCl_3_•6H_2_O) and iron(II) sulfate (FeSO_4_•6H_2_O) were purchased from Merck (Darmstadt, Germany). Chlorophyll a and lutein standards (95% purity) were obtained from Sigma Aldrich (Missouri, MO, USA). About 25% ammonium hydroxide and acetone were obtained from POCH S.A. (Gliwice, Poland). Ethanol was purchased from E.Merck (Darmstadt, Germany). Water with a resistivity of 18.2 MΩ cm was obtained from ULTRAPURE Millipore Direct-Q 3UV-R (Merck, Darmstadt, Germany).

### 4.2. Iron Oxide Nanoparticle (IONP) Synthesis

Iron oxide nanoparticles (IONPs) were obtained using standard co-precipitation under basic conditions in an oxidizing atmosphere without the supply of inert gas. Iron(III) chloride (FeCl_3_) (0.1 M) and iron(II) sulfate (FeSO_4_) (0.1 M) aqueous solutions were mixed in a volume ratio of 2:1, and then 25 mL of 25% ammonia NH_3_ was added dropwise to the solution with constant stirring. The reaction leads to the formation of magnetite (Fe_3_O_4_) in the form of nanoparticles, characterized by stable magnetization at room temperature [77,78,79]. The resulting precipitate was separated using a neodymium magnet and washed several times with deionized water until the solution above the precipitate reached pH 7. The black magnetically active particle precipitate was dried at 45 °C.

### 4.3. Collection of Plant Material and Sample Preparation

The fresh plants *Urtica dioica* L. (Urticaceae) and *Mentha piperita* L. (Lamiaceae) were harvested in the southeastern region of Poland in August 2023. The fresh plants were dried for about three months in a shady area. These samples were then ground into a powder and subjected to extraction. For extraction, 10 g of the plants was suspended in 120 mL of various solvents, i.e., acetone, or 80% ethanol, in a 250 mL Erlenmeyer flask and sonicated for 60 min in an ultrasonic bath (ultrasound power 1200 W, frequency 35 kHz) Bandelin Sonorex RK 103 H (Bandelin Electronics, Berlin, Germany) at 80 °C. After cooling, the extracts were centrifuged at 11,000 rpm for 15 min to precipitate traces of solids from the extract. The supernatants were collected, filtered through Whatman No. 1 filter paper, and evaporated under vacuum. The residue was dissolved in 120 mL of acetone. The obtained samples were refrigerated at 4 °C for further investigations.

### 4.4. MSPE of Photosynthetic Pigments from Plant Extracts

A weighed amount of IONPs was added to 3 mL of acetone extracts of the dried herbs (*Urtica dioica* L., *Mentha piperita* L., or 80% ethanol dry extract dissolved in acetone). The samples were placed in a rotor at maximum speed (30 RPM) to ensure thorough mixing and contact of both phases. Then, after applying a magnet (external field magnetic), phase separation occurred. After phase separation, spectrophotometric measurements were performed for the liquid upper phase using a GENESYS 20 spectrophotometer (Thermo Spectronic, Norristown, PA, USA). Spectra were recorded in the range from 350 nm to 710 nm, and the absorbance measurement for chlorophyll a quantification was performed at 662 nm and for lutein at 450 nm.

### 4.5. The Experimental Adsorption Isotherm

In order to optimize the effectiveness of extraction of chlorophyll a, different mass of IONPs in the range from 0.15 to 0.5 g was added to 3 mL of acetone extract of *Urtica dioica* L. diluted in order to obtain approximately unit absorbance (8-fold dilution using acetone). The tubes were shaken 2 h using a Bio RS-24 Mini Rotator (BioSan, Medical-biological Research and Technologies, Riga, Latvia) at 30 rpm vertical rotation movement (360°). An aliquot of the supernatant was further analyzed spectrophotometrically at 662 nm.

### 4.6. HPLC Analysis

The HPLC analysis was performed on a Merck, Hitachi LaChrom HPLC equipped with a DAD (diode array detector) and a Zorbax Extend C18 Agilent Technologies (Santa Clara, CA, USA) column (150 mm × 4.6 mm I.D., 5 μm). The analysis was performed using gradient elution: 75% acetone (0–5 min), 75–95% acetone (5–10 min), 95% (10–17 min), and 95–100% (17–22 min) at a flow rate of 1.0 mL/min. The detection was carried out at 450 nm for chlorophyll a and 662 nm for lutein.

### 4.7. Quantification of Chlorophyll a and Lutein

#### 4.7.1. HPLC

Stock solutions of 50 mg mL^−1^ chlorophyll a and 1 mg mL^−1^ lutein were prepared in acetone. Stock solutions were serially diluted to the desired concentrations to construct calibration curves with acetone:water mixture (4:1 *v*/*v*). The calibration curves for standards were obtained by plotting the concentration (x, mg mL^−1^) versus peak area (y). The linear regression function were as follows: y = 28,340(±3058.09)x + 46,059(±19,758.17), with *n* = 6, R^2^ = 0.9662, s_e_ = 28,796, F = 85.88, over a range of 1.0–12.5 mg mL^−1^ for chlorophyll a; and y = 44,478,432(±1,758,465)x − 2,679,781(±776,488), with *n* = 7, R^2^ = 0.9922, s_e_ = 911,644, F = 639.78, over a range of 0.1–0.7 mg mL^−1^ for lutein; where R^2^ is the correlation coefficient, s_e_ is the standard error of the regression, F is the value of the Fisher test of significance, and n denotes the number of calibration points. The calibration curve was used for back-calculating chlorophyll a and lutein concentration in the supernatant after the extraction procedure. The limit of detection (LOD) and the limit of quantification (LOQ) were determined from the calibration curves of the standards. The LOD was calculated according to the following expression: the standard deviation of the response × 3/the slope of the calibration curve. The LOQ was established by using the following expression: the standard deviation of the response × 10/the slope of the calibration curve. The LOD values were 2.09 and 0.05 mg mL^−1^, whereas the LOQ values were 6.97 and 0.17 mg mL^−1^ for chlorophyll a and lutein, respectively.

#### 4.7.2. Spectrophotometry

The absorbance was measured at 662 nm for chlorophyll a quantification. The calibration curve was constructed from six standard solutions by plotting the absorbance against the nominal concentration of standard. The linear curve of chlorophyll a was obtained at concentrations ranging from 0.05 to 1.0 mg mL^−1^. The equation of the calibration curve was as follows: y = 1.8118(±0.029)x + 0.0322(±0.016), with *n* = 6, R^2^ = 0.9989, s_e_ = 0.025, F = 3693.28. The LOD and LOQ values were 0.026 mg mL^−1^ and 0.088 mg mL^−1^, respectively. The quantitative determination was based on a standard curve or comparison with the absorbance of the standard solution at a concentration within the linear response.

### 4.8. FT-IR/PAS Measurements

Fourier transform photoacoustic infrared spectra (FT-IR/PAS) of the studied samples were recorded using an Excalibur FT-IR 3000 MX spectrometer (Bio-Rad, Hercules, CA, USA) equipped with a PA301 photoacoustic cell (Gasera, Turku, Finland) within 3800–500 cm^–1^ range, at a resolution of 4 cm^–1^ and 2.5 kHz mirror velocity. Before data collection, dry helium was used to purge the photoacoustic cell. The carbon black standard was used as a reference material to collect FT-IR/PAS spectra. The spectrum consisted of 1024 scans, which provided a good signal-to-noise ratio. No smoothing functions were applied. All spectral measurements were performed at least in triplicate.

### 4.9. XPS Measurements

Powdered samples for XPS analyses were pressed into pellets using hydraulic press. After degassing in load lock for 16 h, they were transferred into the analysis chamber of the Prevac UHV system (Rogów, Poland). All the measurements were carried out using a Scienta R4000 analyzer equipped with a monochromatic (XM 650, 0.2 eV band) Al Kα source (SAX-100, 1486.6 eV, 15 mA, 15 kV). Instrument base pressure was 5 × 10^−9^ mbar. For spectra processing and composition calculations, Casa XPS v. 2.3.25 PR1 software was used.

### 4.10. Nitrogen Adsorption/Desorption Isotherm

Analysis was carried out using an ASAP 2020 instrument (Micromeritics, Norcross, GA, USA). Before the analysis, the sample underwent two degassing cycles under high vacuum conditions. Firstly, in the degassing port, the degassing process lasted 12 h in a temperature of 200 °C. Second, degassing process was conducted in the analysis port just before the proper analysis (4 h in temperature 200 °C). MicroActive V 4.06 software (Micromeritics Norcross, GA, USA) was employed to compute the specific surface area (SBET) and the total pore volume (Vt) from the adsorption/desorption isotherms. The determination of SBET values utilized the standard Brunauer–Emmett–Teller (BET) equation, applied to nitrogen adsorption data within the relative pressure range of P/P_0_ from 0.05 to 0.30. On the other hand, Vt was assessed based on the volume adsorbed at a relative pressure of P/P_0_ of approximately 0.98.

### 4.11. Mathematical Modeling

The equilibrium adsorption isotherm was modeled by the Langmuir equation in the following form:*q*_e_ = *q*_m_ × *K*_L_ × *C*_e_/(1 + *K*_L_ *C*_e_),(1)
where *q*_e_ is the amount adsorbed at the equilibrium conditions, corresponding to the sorbate concentration in the solution (*C*_e_), *K*_L_ is the Langmuir constant, and *q*_m_ is the monolayer capacity, i.e., the maximum amount of sorbate that can be adsorbed under given conditions. The data points (*C*_e_, *q*_e_) were retrieved from measured concentration by using the mass–balance relationship and known operating conditions (mass of the sorbent and volume of the solution).

### 4.12. Molecular Dynamics Simulations

The three systems under consideration included a single solute molecule (chlorophyll a, lutein, or β-carotene) solvated in an explicit solvent, i.e., either water or acetone. Parameters for all solute molecules as well as for acetone corresponded to the GROMOS united-atom force field [80] and were generated by using the Automated Topology Builder online server [81,82]. For water, the SPC model [83] was applied. The sizes of simulation boxes were equal to ca. 5 × 5 × 5 nm^3^ (water-containing systems) or ca. 7.8 × 7.8 × 7.8 nm^3^ (acetone-containing systems). The periodic boundary conditions were applied in all cases. After the initial procedures of geometry minimization and equilibration, the systems were subjected to molecular dynamics (MD) production simulations lasting 50 ns with the data saved every 2 ps. MD simulations were conducted by using GROMACS 2023 package [84] with a timestep of 2 fs. The MD protocol involved the use of the Parrinello–Rahman barostat [85] and the V-rescale thermostat [86]. The solute bond lengths were controlled by using the LINCS algorithm [87], and the non-bonded interactions within the system were treated with a single cutoff distance set to 1.4 nm and the Verlet list scheme. The reaction-field correction [88] was applied using a relative dielectric permittivity of 61.

## 5. Conclusions

This research presents the application of co-precipitation-synthesized IONPs for the removal of chlorophyll from green plant extracts and the concurrent isolation of lutein. The proposed method offers an efficient, versatile, and cost-effective approach for obtaining photosynthetic pigments suitable for various applications. It is not without significance that IONPs are considered safe for humans and the environment.

Characterization of the IONPs’ chemical composition was conducted using SEM-EDS, XPS, and FT-IR/PAS, confirming the sorption of chlorophyll a on the nanoparticles through interactions of its carbonyl groups and FeOOH or FeO on the bio-nano-sorbent surface. This study concludes that an optimal combination of IONPs and acetone extract in a 1:3 ratio (mg chlorophyll a per g IONPs) at room temperature, with a contact time of 48 h for chlorophyll a removal or 72 h for lutein isolation, ensures effective outcomes. Moreover, acetone without the addition of water as a reaction medium is identified as the most effective for the selective sorption of chlorophyll a in the monomeric form. Thanks to the proposed procedure of using IONPs for the purpose of dispersive magnetic extraction into the solid phase of acetone extracts of green plants, on the example of mint and nettle, it is possible to obtain the following benefits: (i) dechlorophyllization of plant extracts, (ii) obtaining IONPs modified with chlorophyll, and (iii) lutein isolation. All stages are cheap, are simple, and do not require advanced equipment and can be used on a larger industrial scale.

## 6. Patents

Polish patent (P.447168, 19 December 2023): “Method for selective removal of chlorophyll a from the extract of green plants and isolation of lutein from the extract of green plants”.

## Figures and Tables

**Figure 1 ijms-25-03152-f001:**
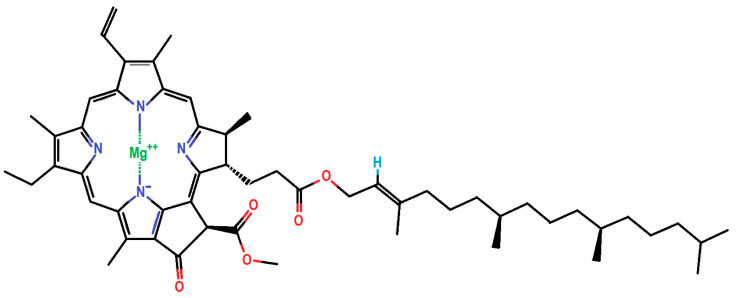
Chemical structure of chlorophyll a based on the PubChem database [40].

**Figure 2 ijms-25-03152-f002:**
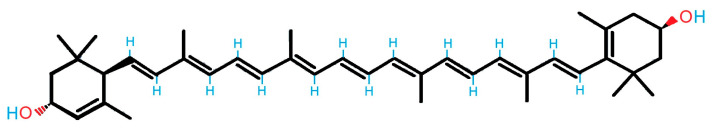
Chemical structure of lutein based on the PubChem database [41].

**Figure 3 ijms-25-03152-f003:**
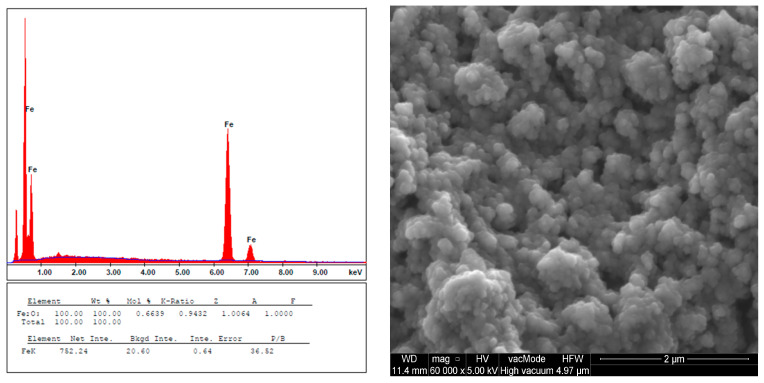
SEM-EDS spectra recorded from IONPs.

**Figure 4 ijms-25-03152-f004:**
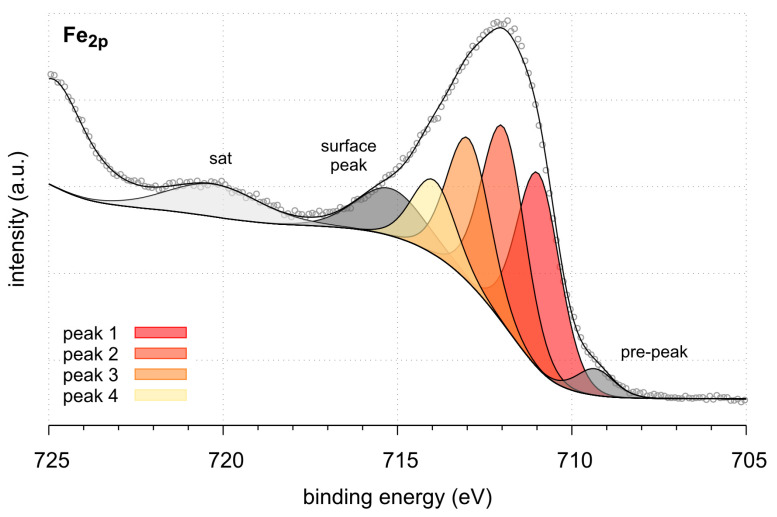
Fe2p spectrum fitted using peak set for FeOOH.

**Figure 5 ijms-25-03152-f005:**
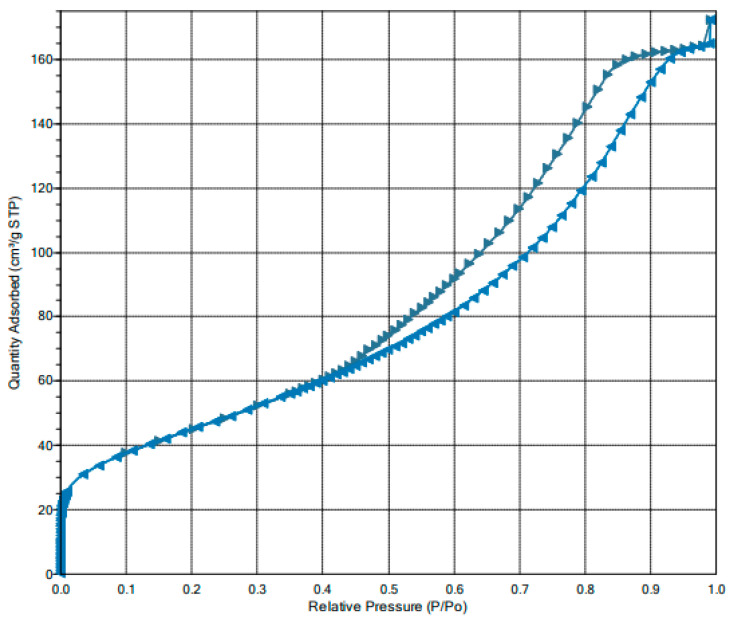
Nitrogen isotherm adsorption–desorption curves of IONPs. Abbreviations: Standard temperature and pressure (STP), *P* and *P*_0_ are the equilibrium and saturation pressure of nitrogen, respectively, the direction of the arrow to the right corresponds to the adsorption process, the direction to the left corresponds to the desorption process.

**Figure 6 ijms-25-03152-f006:**
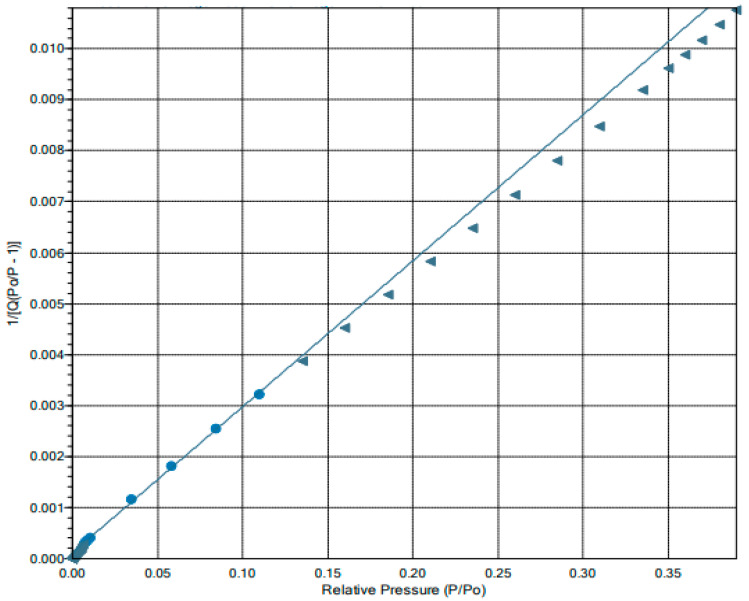
The linear form of BET adsorption isotherm BET plots of IONPs is the ratio of partial pressure of the adsorbed substance to saturated vapor pressure of the adsorbed gas. Points fitted to a straight line are marked with circles, not fitted points are marked with triangles.

**Figure 7 ijms-25-03152-f007:**
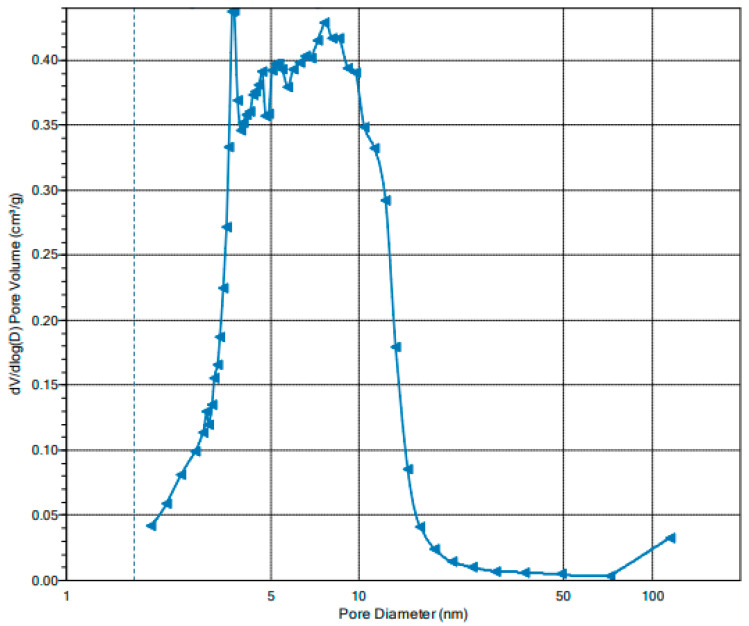
BJH desorption curve of IONPs.

**Figure 8 ijms-25-03152-f008:**
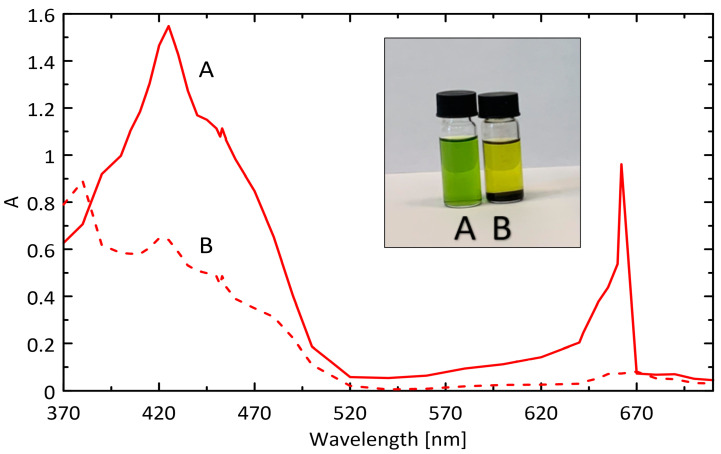
Spectrum of the acetone extract of *Urtica dioica* L. (solid line) diluted 8 times with acetone and the spectrum obtained after adding 400 mg of IONPs to 3 mL of the diluted extract. Insert illustrates a two-phase system (B, yellow, dot line) created after mixing IONPs and acetone extract of *Urtica dioica* L. (A, green, solid line).

**Figure 9 ijms-25-03152-f009:**
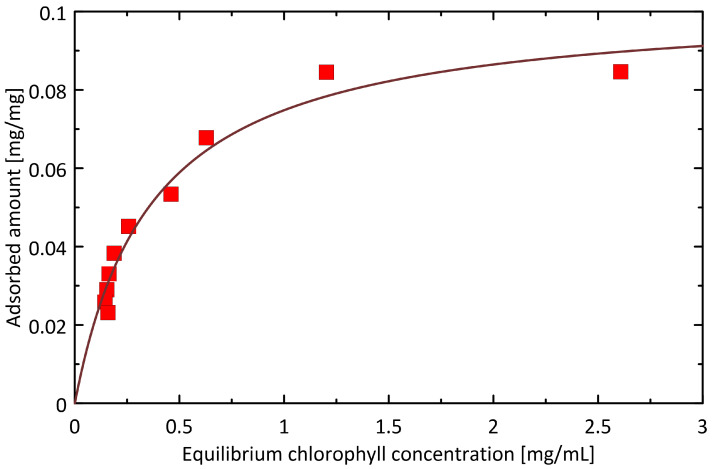
The experimental adsorption isotherm measured for the case of chlorophyll a sorption onto IONPs. The solid line represents the best fit offered by the Langmuir model of adsorption.

**Figure 10 ijms-25-03152-f010:**
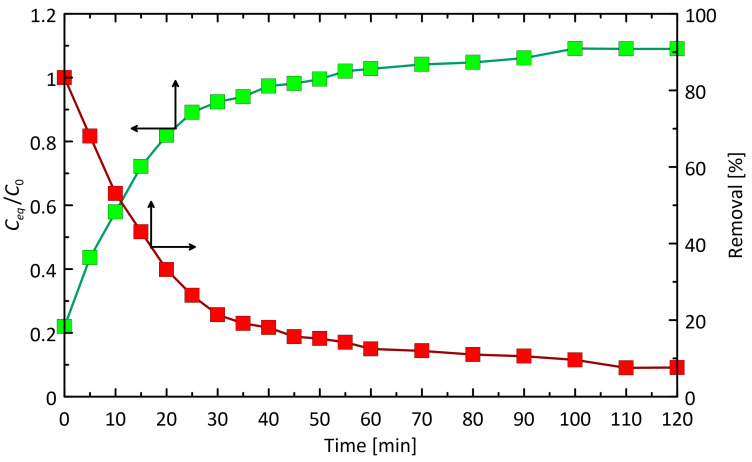
The experimental curve representing the adsorption kinetics of chlorophyll a into the IONPs. Analysis conditions: 400 mg IONPs, extract volume 3 mL after 8-fold dilution, spectrophotometric measurement at a wavelength of 662 nm.

**Figure 11 ijms-25-03152-f011:**
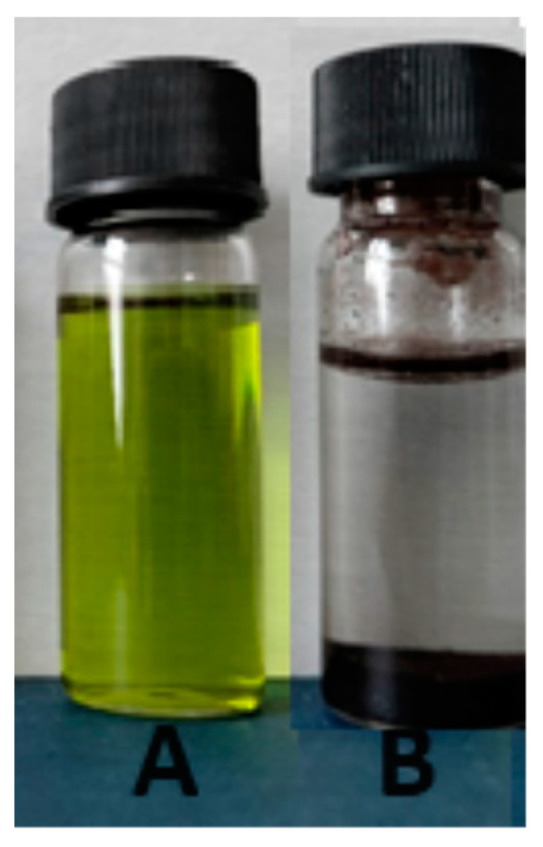
*Urtica dioica* L. extract in acetone (**A**). Extract containing 10% acetone in water and 400 mg MNPs (**B**).

**Figure 12 ijms-25-03152-f012:**
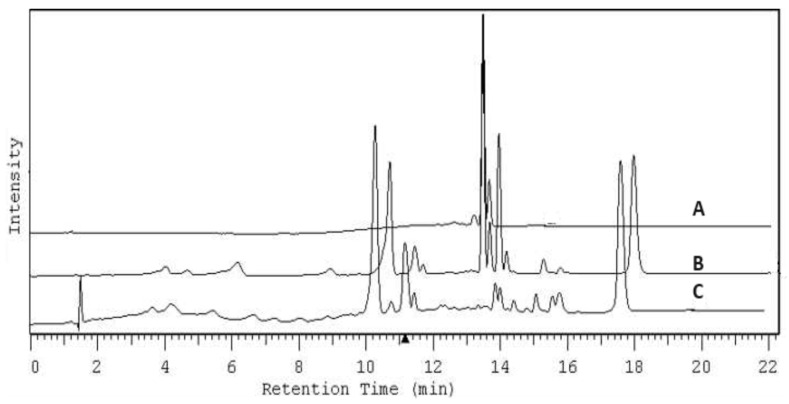
Chromatogram of the chlorophyll a standard with a concentration of 3.65 mg mL^−1^. (A), chromatogram of the acetone extract from *Urtica dioica* L. (B), and chromatogram after 48 h of contact with IONPs (C).

**Figure 13 ijms-25-03152-f013:**
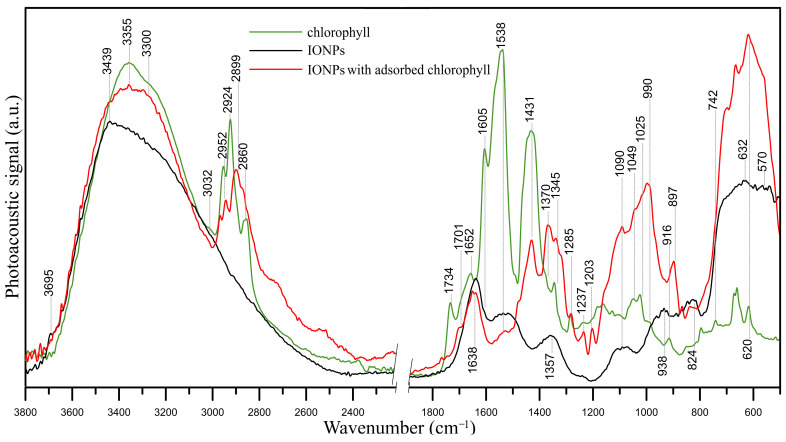
FT-IR/PAS spectra of chlorophyll (green line), IONPs (black line), and IONPs with adsorbed chlorophyll (red line).

**Figure 14 ijms-25-03152-f014:**
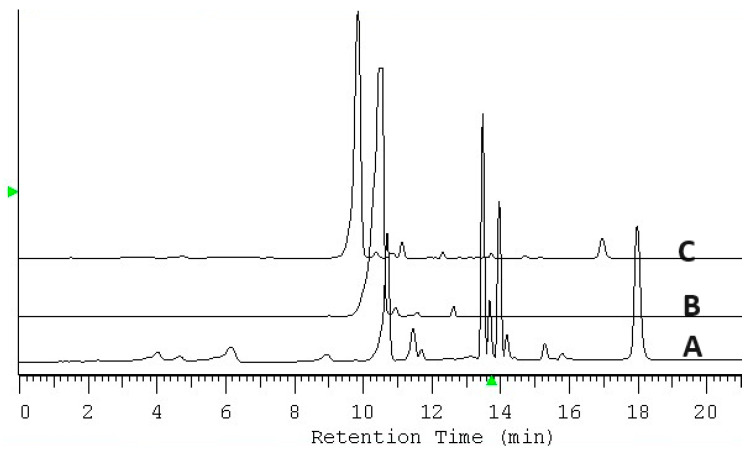
Chromatogram of the acetone extract from *Urtica dioica* L. (A), chromatogram of the lutein standard with a concentration of 0.57 mg mL^−1^ (B), and chromatogram of the extract after 72 h of contact time with IONP (C). Green point indicates chlorophyll retention.

**Figure 15 ijms-25-03152-f015:**
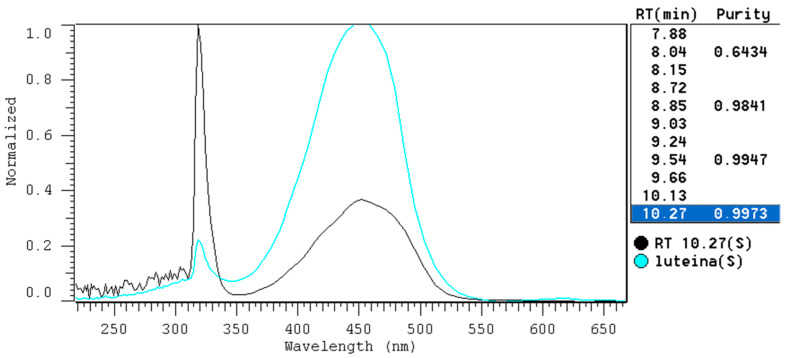
Overlapped spectra, recorded by a DAD, for the lutein standard (lutein S) and isolated lutein (RT 10.27 S).

**Table 1 ijms-25-03152-t001:** The results of fitting of Fe_2p_ spectrum with FeOOH peaks: binding energies (BE), peak widths (eV), and their intensities (%).

Sample	Peak	BE (eV)	FWHM (eV)	Intensity (%)
Fe-nps	FeOOH peak 1	711.0	1.47	25.1
	FeOOH peak 2	712.0	1.47	25.3
	FeOOH peak 3	713.0	1.47	18.5
	FeOOH peak 4	714.0	1.47	9.6
	pre-peak	709.3	1.27	3.0
	surface peak	715.2	2.30	9.5
	sat 1	720.3	3.18	9.0

**Table 2 ijms-25-03152-t002:** The N_2_ isotherm adsorption/desorption experimental results.

Method	Constant	Quantity in the Monolayer	Surface Area	Molecular Cross-Sectional Area	Correlation Coefficient
Langmuir	106.638 1/mmHg	19.291 cm^3^ g^−1^ STP ^1^	83.963 ± 0.858 m^2^ g^−1^	0.1620 nm^2^	0.9984
BET	256.731	34.779 cm^3^ g^−1^ STP ^1^	151.377 ± 1704 m^2^ g^−1^	0.1620 nm^2^	0.9996

^1^ Standard temperature and pressure (STP).

**Table 3 ijms-25-03152-t003:** IONP pore structure parameters calculated using the Barrett–Joyner–Halenda (BJH) method.

Cumulative Surface Area of Pores		Cumulative Volume of Pores		Average Diameter of Pores	
Adsorption	Desorption	Adsorption	Desorption	Adsorption	Desorption
143.947 m^2^ g^−1^	181.145 m^2^ g^−1^	0.2373 cm^3^ g^−1^	0.2715 cm^3^ g^−1^	6.5942 nm	5.9961 nm

**Table 4 ijms-25-03152-t004:** Effect of water on the sorption of photosynthetic pigments from *Urtica dioica* L extract on IONPs. Spectrophotometric measurements were performed at 662 nm and 452 nm before and after the addition of IONPs.

Time	ΔA [%]	% Acetone in Water									
		100%	90%	80%	70%	60%	50%	40%	30%	20%	10%
10 min	662 nm	34.75	2.73	5.42	3.05	4.39	50.00	79.44	81.11	78.29	75.42
	452 nm	40.69	0.16	0.21	0.35	17.93	76.99	76.07	73.99	51.86	56.53
20 min	662 nm	59.17	0.11	−3.62	5.91	55.34	88.08	90.56	85.46	90.36	81.99
	452 nm	44.39	0.05	1.86	3.68	30.18	83.59	81.77	78.03	68.22	66.22
60 min	662 nm	84.65	1.53	−0.11	47.09	70.43	93.35	92.50	92.73	95.25	98.48
	452 nm	56.81	3.62	15.14	20.07	60.79	91.49	86.34	90.20	87.54	87.06
120 min	662 nm	90.62	−1.03	−0.21	47.95	89.83	79.44	81.11	78.29	75.42	64.34
	452 nm	58.39	5.71	13.48	21.56	87.62	94.93	88.06	92.52	94.97	95.48

**Table 5 ijms-25-03152-t005:** The average energies of interactions between sorbate molecules and solvent (water or acetone) as retrieved from unbiased MD simulations (in (kJ/mol)). The energies are split into Lennard–Jones (LJ) and Coulombic contributions.

Solute	Water			Acetone			Difference
	Coulomb	LJ	Sum	Coulomb	LJ	Sum	
Chlorophyll	−106.3	−259.9	−366.2	−73.7	−469	−542.7	−176.5
Carotene	−31.2	−239.1	−270.3	−18.5	−331.8	−350.3	−80
Lutein	−60.8	−244	−304.8	−38	−345.7	−383.7	−78.9

## Data Availability

Data will be made available on request from J. Flieger.

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
