# Peer review of "Selective Removal of Chlorophyll and Isolation of Lutein from Plant Extracts Using Magnetic Solid Phase Extraction with Iron Oxide Nanoparticles"

_ijms, 2024, doi:10.3390/ijms25063152_

Round 1

Reviewer 1 Report

Comments and Suggestions for Authors

-The keywords should be revised according to the journal writing rules and should be rearranged in alphabetical order

Table 3, I saw a reference in the title, this is not a correct usage, if a reference is to be used, it is referenced to the model on which it is based in the text

The discussion part is quite weak, it should be improved

The conclusion section should be rewritten, the striking results should be highlighted in this section, and finally, suggestions should be included

Especially the figure captions and superscripts, and the font sizes used in the figure are all different from each other, they should be given in a uniform format

Units for the text should be written uniformly

Anti-cancer (replace all text)

introductory part is not enough, it is not clearly stated why this study was done, why this study was needed. should be discussed in the light of previous studies, deficiencies should be criticized

Author Response

The authors would like to thank the reviewer for taking the time to read our work. Every critical comment is valuable to us and has been used in the preparation of a new version of the work. We tried to modify the discussion, introduction, and conclusion to clarify our intentions and comment on the obtained result.

  1. The keywords should be revised according to the journal writing rules and should be rearranged in alphabetical order

Answer: Thank You for this suggestion. We did it.

  1. Table 3, I saw a reference in the title, this is not a correct usage, if a reference is to be used, it is referenced to the model on which it is based in the text

Answer: Thank You for this suggestion. We agree with the reviewer. The table, however, was transferred to supplementary materials in the new version.

  1. The discussion part is quite weak, it should be improved

Answer: The discussion part has been improved. We do hope that now is better.

  1. The conclusion section should be rewritten, the striking results should be highlighted in this section, and finally, suggestions should be included

Answer: Suggestions for future investigations have been added to the Discussion part. The conclusion has been improved.

  1. Especially the figure captions and superscripts, and the font sizes used in the figure are all different from each other, they should be given in a uniform format

Answer: Yes we agree with the reviewer. Thank you for drawing our attention to this aspect. We corrected Fig.8 intending to make a uniform legend description, in Fig.5,3 we did again using GraphPad. Unfortunately, the remaining figures are created by instruments SEM-EDS, FTIR, and HPLC.

  1. Units for the text should be written uniformly

Answer: The units were checked and improved to be uniform.

  1. Anti-cancer (replace all text)

Answer: This part was removed from the Discussion.

  1. introductory part is not enough, it is not clearly stated why this study was done, or why this study was needed. should be discussed in the light of previous studies, deficiencies should be criticized

Answer: Yes. Thank You for this suggestion. We did our best to clarify the above issues.

Reviewer 2 Report

Comments and Suggestions for Authors

The reviewed paper is devoted to the isolation of important dyes by using an advanced MSPE technology. Authors from Poland follow initial experiments of Michael Tswett, who reported the discovery of chromatography at the meeting of the Warsaw Society of Natural Sciences in 1903. Fine iron oxide powder was tested by him as one of many potentially useful adsorption materials for the separation of chlorophylls. May be it would not be worth to mention this fact. Anyway, the paper is well written and contains new useful data on the extraction of pigments, so it can be recommended for the publication in the International Journal of Molecular Sciences after some revision and addition of missing characteristics of IONP.

Probably, the choice of Mentha piperita L., Urtica dioica L. is not optimal. The former is an important and valuable material for the production of essential oils. The latter is known as a stinger, so its collection is difficult. It would be useful to justify the choice and propose possible alternative plants. May be, it would be useful to consider the possibility of the combination of two processes – production of essential oils and extraction of chlorophyll a in the case of Mentha piperita L.

The paper is big with too detailed description of the experiments, which is sometimes overloading the main content. For example, XPS characterization of magnetic particles (Section 2.1.2) is presented in 4 pages (pp.3-6) containing exhaustive 3(!) tables and 3(!) figures. Certainly, this section is not a key part of this article and it should be condensed without any damage to the content. Some part of the corresponding results can be transferred to Supporting Information.

It is well known that magnetic properties of magnetite depend on particle size of IONP. However, the corresponding data are missing in the article.

Also, it would be useful to measure the specific surface area and pore size of IONP aggregates to understand better the kinetics of the sorption and isotherm of adsorption.

Author Response

The reviewed paper is devoted to the isolation of important dyes by using an advanced MSPE technology. Authors from Poland follow initial experiments of Michael Tswett, who reported the discovery of chromatography at the meeting of the Warsaw Society of Natural Sciences in 1903. Fine iron oxide powder was tested by him as one of many potentially useful adsorption materials for the separation of chlorophylls. May be it would not be worth to mention this fact. Anyway, the paper is well written and contains new useful data on the extraction of pigments, so it can be recommended for the publication in the International Journal of Molecular Sciences after some revision and addition of missing characteristics of IONP.

Answer: We thank the reviewer for his contribution to improving our manuscript. We appreciate the effort and time devoted to our work. We have tried to improve the manuscript in accordance with your comments.

Probably, the choice of Mentha piperita L., Urtica dioica L. is not optimal. The former is an important and valuable material for the production of essential oils. The latter is known as a stinger, so its collection is difficult. It would be useful to justify the choice and propose possible alternative plants. May be, it would be useful to consider the possibility of the combination of two processes – production of essential oils and extraction of chlorophyll a in the case of Mentha piperita L.

Answer: Thank you for this suggestion. Indeed, we can repeat the experiment using different green plants. We chose two popular plants as examples to expand the applicability of the patented method. We included these two plants in the work because the results are very similar in other cases we examined for it to bring anything new to the work. The idea to also analyze essential oils is very interesting. New experiments and methods would have to be planned. We hope to use this in future work or inspire other authors, so this issue has been added to the future perspectives discussion section.

The paper is big with too detailed description of the experiments, which is sometimes overloading the main content. For example, XPS characterization of magnetic particles (Section 2.1.2) is presented in 4 pages (pp.3-6) containing exhaustive 3(!) tables and 3(!) figures. Certainly, this section is not a key part of this article and it should be condensed without any damage to the content. Some part of the corresponding results can be transferred to Supporting Information.

Answer: Yes. Thank you for this suggestion We agree with the reviewer. We prepared supplementary information to this manuscript and transferred two figures and two tables to this part of the submission.

It is well known that magnetic properties of magnetite depend on particle size of IONP. However, the corresponding data are missing in the article. Also, it would be useful to measure the specific surface area and pore size of IONP aggregates to understand better the kinetics of the sorption and isotherm of adsorption.

Answer: Yes, it is a very important subject. Thank you for drawing our attention to these aspects. We have recently been studying iron oxide nanoparticles and are gradually supplementing our research with new instrumental methods.Magnetism issue we planned to add to our next paper we are preparing to Molecules currently. However, we have added explanatory text and the above issue as the weaknesses of the work and the prospect of further research. Taking into account the reviewer's suggestion we added a new subchapter 2.2 devoted to the specific surface area and pore size of IONPs.

Reviewer 3 Report

Comments and Suggestions for Authors

I am very grateful you for the invitation to review the manuscript ijms-2891963 by Flieger and coauthors Selective removal of chlorophyll and isolation of lutein from plant extracts using magnetic solid phase extraction with iron oxide nanoparticles”. The work is interesting but needs adjustments to increase the quality of the material.

Comments:

·       iThenticate found a similarity of 16% in this work. A maximum of 10% is recommended for publication. This needs to be reviewed, it is essential

·       Authors should focus more on the results and conclusions obtained rather than the introduction and method in the abstract.

·       Why was there no mention of lutein, stating why they studied the isolation of this compound? The authors should mention this.

·       Why was acetone chosen as the extracting agent? We know how dangerous these volatile organic compounds are in the environment. Nowadays there are so many alternative solvents for this type of organic solvent. The choice of acetone has to be justified, especially as numerous studies using these types of solvents have already been reported in the literature.

Author Response

I am very grateful you for the invitation to review the manuscript ijms-2891963 by Flieger and coauthors “Selective removal of chlorophyll and isolation of lutein from plant extracts using magnetic solid phase extraction with iron oxide nanoparticles”. The work is interesting but needs adjustments to increase the quality of the material.

 Answer: We thank the reviewer for his contribution to improving our manuscript. We appreciate the effort and time devoted to our work. We have tried to improve the manuscript in accordance with your comments.

  • iThenticate found a similarity of 16% in this work. A maximum of 10% is recommended for publication. This needs to be reviewed, it is essential

Answer: Thank you for this suggestion. Unfortunately, we do not have access to this system. If the publishing house provides us with an anti-plagiarism report, we will be able to refer to it and we will know what to improve. Probably, after the current improvement, this value has also changed. We are open to this matter.

  • Authors should focus more on the results and conclusions obtained rather than the introduction and method in the abstract.

Answer: We corrected Discusion and Conlusion. Thank you for this suggestion.

  • Why was there no mention of lutein, stating why they studied the isolation of this compound? The authors should mention this.

Answer: We added information about lutein together with its structure.

  • Why was acetone chosen as the extracting agent? We know how dangerous these volatile organic compounds are in the environment. Nowadays there are so many alternative solvents for this type of organic solvent. The choice of acetone has to be justified, especially as numerous studies using these types of solvents have already been reported in the literature.

Answer: We agree with the reviewer that it would be good to replace acetone. Our experience with sorption on IONPs shows that only it can provide selective sorption. Only a small amount of water (up to 30%) can be added. Sorption from aqueous solutions is efficient but non-selective. Most articles applying IONPs present sorption under aqueous conditions. We have some practice using ionic liquids and DES in extraction. They were successfully used to isolate heme/hemoglobin, which has a chemical structure similar to chlorophyll. So there is a lot of potential in this idea. However, these are topics for another work.

Round 2

Reviewer 1 Report

Comments and Suggestions for Authors

congratulations

Reviewer 2 Report

Comments and Suggestions for Authors

The authors of the manuscript met all points raised by me, so I am happy to recommend the revised version for the publication in IJMS in the current form.